# Towards a Sustainable and Safe Future: Mapping Bike Accidents in Urbanized Context

Ahmed Jaber * and Bálint Csonka

Department of Transport Technology and Economics, Faculty of Transportation Engineering and Vehicle Engineering, Budapest University of Technology and Economics, Műegyetem rkp. 3., H-1111 Budapest, Hungary; csonka.balint@kjk.bme.hu
* Correspondence: ahjaber6@edu.bme.hu

**Abstract:** This manuscript presents a study on the spatial relationships between bike accidents, the built environment, land use, and transportation network characteristics in Budapest, Hungary using geographic weighted regression (GWR). The sample period includes bike crash data between 2017 and 2022. The findings provide insights into the spatial distribution of bike crashes and their severity, which can be useful for designing targeted interventions to improve bike safety in Budapest and be useful for policymakers and city planners in developing effective strategies to reduce the severity of bike crashes in urban areas. The study reveals that built environment features, such as traffic signals, road crossings, and bus stops, are positively correlated with the bike crash index, particularly in the inner areas of the city. However, traffic signals have a negative correlation with the bike crash index in the suburbs, where they may contribute to making roads safer for cyclists. The study also shows that commercial activity and PT stops have a higher impact on bike crashes in the northern and western districts. GWR analysis further suggests that one-way roads and higher speed limits are associated with more severe bike crashes, while green and recreational areas are generally safer for cyclists. Future research should be focused on the traffic volume and bike trips' effects on the severity index.

**Keywords:** micro-mobility; bike crashes; spatial analysis; accidents; safety; cycling





## 1. Introduction

Due to its many positive effects on individual, public, and environmental health [1], cycling has become a common form of urban transportation in many cities around the globe [2,3]. However, cycling can also be risky, especially in congested urban areas where collisions with cars are frequent. City planners and policymakers can design safer transportation systems that prioritize the safety of all road users [4]. This requires a deeper understanding of the factors affecting cycling crashes [5].

Many factors, including the built environment, land use, and transportation network characteristics, influence cycling accidents [6–8]. Cyclists are at a higher risk of injury or death on roads with higher speed limits, wider lanes, and heavier traffic [9,10]. Roundabouts, intersections, and bike lanes may affect the frequency of accidents involving cyclists [11,12]. When there are no bike lanes at an intersection, for instance, cyclists must weave in and out of traffic to make turns, which can be dangerous. Land use also significantly influences cyclists' safety; bicycle accidents are more likely to happen in areas with a high population density, a variety of land uses, and lots of pedestrian traffic [13,14]. Furthermore, areas with high concentrations of cyclists and motorists can result in conflicts that lead to accidents [15,16]. Considering these threats when planning cities and neighborhoods helps to lessen the negative impact of cycling. Cities can encourage healthy and sustainable lifestyles by prioritizing cycling infrastructure and land-use patterns that support active transportation and reduce the number of accidents involving cyclists.

The main contribution of this research is the definition of the spatial relationships between the bike crashes and the built environment, land use, and transportation network characteristics, which can have significant impacts on frequency and severity. The novelty of this study is that bike crashes have not been spatially studied using geographic weighted regression (GWR) in aspects of land-use patterns, speed limits, road classification, and rail network densities in urbanized area. GWR can provide insights into the spatial patterns of bicycle crashes and help policymakers identify high-risk areas and develop targeted interventions to reduce the severity of crashes.

## 2. Literature Review

In recent years, there has been a growing recognition of cycling safety [17], as well as the crucial role that spatial analysis plays in understanding and addressing this issue [18]. It is a pressing concern within the broader context of urban transportation and road infrastructure.

Several studies have identified various factors that contribute to cycling crashes, including individual factors such as age, gender, and experience level [19–21] and environmental factors such as road conditions and traffic volume [22,23]. Some studies found that elderly people had a higher risk of severe injuries in cycling crashes [24–26]. Other studies identified road design and infrastructure as critical factors in cycling crashes, including the presence of dedicated cycling lanes and intersections [27–29].

Focusing on cycling crashes in intersections, Pulvirenti et al. [30] highlighted the uncertainties surrounding roundabout impacts on bicyclists' safety, emphasizing the complexities of interactions between bicyclists and vehicles. Additionally, studies [31,32] underscored the significance of intersection safety for cyclists, offering methodologies like the Bicycle Intersection Safety Index (Bike ISI) to proactively prioritize safety improvements. Furthermore, the insights of Daniels et al. [33] into roundabout conversions emphasized the role of well-designed cycling infrastructure.

Traffic-calming measures and adequate lighting can significantly reduce the risk of cycling crashes [34–36]. On the other hand, poor road conditions, such as potholes, uneven surfaces, and inadequate signage, can increase the risk of cycling crashes [37–39]. In addition, the presence of barriers, such as high curbs or fences, can make it difficult for cyclists to navigate the road safely [40,41]. Designing cycling infrastructure that prioritizes the safety and needs of cyclists can reduce the risk of cycling crashes and make cycling a more accessible and attractive transportation mode.

Other important factor affecting cycling crashes is the behavior of cyclists and drivers. Several studies have found that risky behaviors, such as cycling at high speed, running red lights, and lack of protective gear, can increase the likelihood of cycling crash severity [42–45]. In addition, drivers who are distracted, speeding, or under the influence of drugs or alcohol pose a significant risk to cyclists on the road [46–48]. Education and awareness campaigns aimed at promoting safe cycling and driving behaviors can reduce the risk of cycling accidents [49,50].

Spatial factors also play a crucial role in cycling crashes. Some studies found that the risk of cycling crashes increased in areas with high traffic volume and high speed limits, particularly in urban areas [51]. Other studies also found that areas with high population density and high levels of economic activity were more likely to have cycling crashes, indicating a need for better urban planning and transportation policies [52,53].

Geographically weighted regression (GWR) is a statistical technique that has been increasingly used in transportation research to analyze the spatial patterns of transportation-related phenomena [54–56], including crash-severity analysis. To focus on the cycling-crash-severity analysis based on GWR, we present a summary of the literature discussing this issue.

Previous studies, as shown in in Table 1, demonstrate the utility of GWR in modeling the severity of bicycle crashes. Clear gaps are extracted from the literature, such as specific land-use patterns as the industrial, water, commercial, recreational, and green areas, as

well as the speed limit and the density of road and railway networks, which are the main novelties of our research.

**Table 1.** Related literature review on bike crash analysis using GWR.

| Ref. | Spatial-Related Results | Gaps |
|---|---|---|
| [57] | 1. Bus stop density is associated with cyclist crashes<br>2. Car–cyclist crashes occur in stop-sign-controlled intersections<br>3. More signalized intersections lead to more crashes | Focused only on car–cyclist crashes<br>Did not consider the land-use patterns<br>Did not consider the road speed limits<br>Did not consider the railway network |
| [58] | 1. The more central a network, the safer it is for bicyclists<br>2. Bicycling is safer on major roads than on local roads<br>3. Commercial area properties tend to experience a greater number of bicyclist-involved crashes<br>4. Bus stop density is associated with cyclist crashes | Cyclist facilities, such as bike lots and bike lanes, are not taken into consideration<br>Did not consider the road speed limits<br>Did not consider the railway network |
| [59] | 1. The agricultural area has a significant negative correlation with bicycle crashes<br>2. Collector roads were significant and positively associated with bicycle crashes | Land-use patterns are limited<br>Did not consider the road speed limits<br>Did not consider the railway network<br>Did not consider the road classification |
| [60] | 1. Significant associations between bicycle crashes and bicycle lane intersection density<br>2. Sidewalk density and commercial areas are positively associated with bicycle crashes<br>3. The effect of residential road density, percentage of single-family areas, and percentage of multiple-family areas positively vary across the space | Did not consider the road speed limits<br>Did not considered the railway network<br>Did not consider the road classification |

## 3. Methodology

### 3.1. Framework

GWR is a local regression model that accounts for spatial heterogeneity in the relationship between the dependent variable and the independent variables. The technique involves estimating a separate regression model for each location or point in space, in which the coefficients of the independent variables are allowed to vary spatially. The resulting local models can then be used to explore spatial variation in the relationship between the dependent and independent variables [54,61] and help to identify any spatial heterogeneity in the relationships that may be missed by a global regression model [56,60,62–64]. The framework is summarized as follows:

1. The data should be converted to georeferenced data;
2. An appropriate model is selected, such as linear regression, logistic regression, or Poisson regression;
3. To determine the influence of nearby data points on the regression estimation at a given location, the kernel function is used, which is a weighting function. The most used kernel functions are Gaussian, bi-square, and exponential;
4. The optimal bandwidth parameter for the GWR model should be selected to determine the size of the local neighborhood that is used to estimate the regression coefficients. This is to consider the extent of the spatial autocorrelation and to avoid overfitting or underfitting the model;
5. The GWR model is fitted using the selected model and bandwidth parameters. This involves estimating the regression coefficients for each location in the study area. This is carried out by applying the kernel function to each data point within the local neighborhood and solving the resulting weighted least-squares regression problem;
6. Model validation, such as cross-validation, residual analysis, or goodness-of-fit statistics, is carried out;
7. The GWR coefficients' maps are created to visualize spatial patterns.

Overall, GWR is a powerful technique for analyzing spatial data that can provide valuable insights into the spatial variation in the relationships between variables. It is

important to carefully select the appropriate model and bandwidth parameters, validate the model, and interpret the results in the context of the study area.

### 3.2. Data Source and Study Area

Budapest, Hungary is an urban area with a population density of more than 3000 people per square kilometer. The river Danube separates Budapest into east and west sides. All variables of the study area were cataloged in a geographic information system (GIS) database alongside crash incidents. The database includes bike crash data between 2017 and 2022 provided by the Centre for Budapest Transport (Budapesti Közlekedési Központ; BKK). The datasheet has 2682 crashes (2796 injuries), including 9 fatalities (0.32%), 736 serious injuries (26.32%), and 2051 slight injuries (73.36%). Each crash has its geographic position in latitude and longitude coordinates. The dataset consists only of the location and the severity of the crashes. The other geographic information about road network, railway network, facilities, stops, road classification, speed limit, intersections, signals, and land use within the study area were obtained from OpenStreetMap. A 1 × 1 km square grid cell was used because it provides a greater resolution of data than sub-districts and because it is recommended in the literature [54,65,66]. These cells are referred to in the manuscript as "zones". The total number of cells/zones is 1274. The collected data were assigned to the appropriate cells.

## 4. Data and Model Preparation

### 4.1. Variables

This study uses the bike-crash-severity index as a dependent variable. Three severity categories were distinguished, and 1, 3, and 5 points were assigned to slight, serious, and fatal injuries, respectively. For each zone, the severity index values were summarized (1). Thus, the higher the bike-crash-severity index is, the riskier the zone is. The rationale behind our use of the bike-crash-severity index lies in its ability to succinctly capture the varying degrees of injuries resulting from bike crashes, providing a comprehensive measure of the overall impact on cyclist safety within each zone. The assignment of 1, 3, and 5 points to slight, serious, and fatal injuries, respectively, has been established with careful consideration of the relative impact of these injury categories on cyclist safety. Although there are some alternative weighting approaches [67,68], our intention is to create a simplified yet meaningful severity index that aligns with the objective of identifying zones with higher risk levels.

$$i = n_{slight} + 3n_{serious} + 5n_{fatal} \tag{1}$$

where $n_{slight}$, $n_{serious}$, and $n_{fatal}$ are the number of injuries in the severity categories. The descriptive statistics of variables are shown in Table 2. The data show that the bike crash index is between 0 and 109.

### 4.2. Data Cleaning and Model Characteristics

Multicollinearity among variables, such as unstable coefficients and inflated standard errors, can cause problems in the model and should be removed. One way to detect multicollinearity is to use the variance inflation factor (VIF), which measures the extent to which the variance of the estimated regression coefficient for a predictor variable is increased due to multicollinearity with the other predictor variables in the model. VIF values greater than 7.5 indicate problematic levels of multicollinearity [64]. Table 3 shows the VIF values for all variables. Residential areas, two-way roads, and roads with a speed limit > 100 km/h variables have VIF values greater than 7.5 and are eliminated from the GWR model.

**Table 2.** Descriptive statistics of the variables.

| Category | Variable | Average | Minimum | Maximum |
|---|---|---|---|---|
| Built Environment and Transportation Facilities (number/zone) | Touristic Points | 2.01 | 0 | 81 |
| | Crossings | 6.54 | 0 | 96 |
| | Traffic Signals | 1.89 | 0 | 41 |
| | Rail Stops | 0.60 | 0 | 11 |
| | Bus Stops | 3.56 | 0 | 36 |
| | Bike Lots | 1.11 | 0 | 45 |
| Road Speed Limit and Regulations (km/zone) | One-way | 1.47 | 0 | 12.45 |
| | Two-way | 9.83 | 0 | 38.91 |
| | Speed Limit $\leq$ 30 km/h | 1.86 | 0 | 11.37 |
| | Speed Limit 31–50 km/h | 1.06 | 0 | 12.38 |
| | Speed Limit 51–100 km/h | 0.29 | 0 | 4.27 |
| | Speed Limit > 100 km/h | 0.07 | 0 | 2.63 |
| Road Functions (km/zone) | Bike Road | 0.52 | 0 | 9.90 |
| | Pedestrian Street | 2.01 | 0 | 24.32 |
| | Highway | 0.14 | 0 | 8.49 |
| | Residential Road | 3.54 | 0 | 12.44 |
| | Main Road | 0.67 | 0 | 7.71 |
| | Service Road | 2.00 | 0 | 16.19 |
| Railway service (km/zone) | Tram | 0.38 | 0 | 11.06 |
| | Train | 0.99 | 0 | 42.71 |
| | Light Rail | 0.11 | 0 | 7.65 |
| | Subway | 0.12 | 0 | 16.86 |
| Land Use (%/zone) | Commercial | 1.93% | 0 | 97% |
| | Industrial | 9.64% | 0 | 100% |
| | Water | 3.76% | 0 | 100% |
| | Green | 37.87% | 0 | 100% |
| | Recreation | 0.21% | 0 | 64% |
| | Residential | 44.67% | 0 | 100% |
| Crashes | Severity index | 3.32 | 0 | 109 |

**Table 3.** VIF values of the variables.

| Variable | VIF | Variable | VIF | Variable | VIF |
|---|---|---|---|---|---|
| Touristic Points | 2.62 | Water Area | 1.36 | Cycling Road | 1.38 |
| Crossings | 4.39 | Green Area | 3.65 | Pedestrian Road | 2.51 |
| Traffic Signals | 4.21 | Recreation Area | 1.02 | Highway | 1.79 |
| Rails Stops | 4.40 | Residential Area | 8.36 * | Residential Road | 4.49 |
| Bus Stops | 2.60 | Two-way | 10.4 * | Main Road | 3.42 |
| Bike Lots | 3.38 | One-way | 5.00 | Service Road | 1.96 |
| Commercial Area | 1.22 | Speed Limit $\leq$ 30 km/h | 1.97 | Tram way | 3.76 |
| Industrial Area | 1.92 | Speed Limit 31–50 km/h | 2.02 | Rail way | 1.28 |
| | | Speed Limit 51–100 km/h | 2.32 | Light rail way | 1.14 |
| | | Speed Limit > 100 km/h | 7.93 * | Subway | 1.24 |

* VIF values are greater than 7.5 and eliminated from the GWR model.

GWR relies on a kernel function with a specific bandwidth. There are two kinds of kernels that can be used with GWR: the fixed kernel chooses neighbors on a distance threshold and the adaptive kernel chooses neighbors based on predetermined number of neighbors. Since the size of the block groups is constant throughout this investigation, the fixed kernel is selected. In GWR, the selection occurs from one of these methods: Akaike information criterion (AICc), cross-validation (CV), or bandwidth parameter. In this research, we use the cross-validation technique to implement the GWR model. Cross-validation (CV) employs a bandwidth to identify a model that minimizes the difference between the observed and fitted values. For the GWR analysis, the optimal distance is determined by two factors: the randomness of the GWR residuals and a lower AICc value compared to the other bandwidth models. ArcMap's best model can be obtained through

an automated function. The adjusted R-squared values show that 78.6% of the variation in the bike crash index is explained by independent variables, which is significantly higher than the 57.2% explained by the ordinary least-squares method. With an AICc of 7488.25, the best-applied bandwidth is 8027 m.

As a summary, the model development steps are as follows: the methodology aim is to comprehensively address the spatial aspects of cyclist safety at road intersections. We initiated the process by defining the research objectives, aiming to investigate safety patterns and risks. Data collection involved sourcing infrastructure details from OpenStreetMap and bike crash data from the Centre for Budapest Transport. To quantify bike crash severity, we introduced a bike-crash-severity index, categorizing injuries into slight, serious, and fatal categories and assigning corresponding points. Geographic information systems (GISs) are pivotal in our analysis, enabling us to overlay infrastructure data and severity indices, thereby identifying high-risk zones and visualizing spatial patterns. Employing geographically weighted regression, we explore the localized relationships between infrastructure variables and bike crash severity.

## 5. Results and Discussion

Figures 1–5 depict the model's contribution to the coefficients of statistically significant variables, with dark red denoting the highest values and light red indicating the lowest. It should be noted that the term "positive" means positive sign of the coefficients, which indicates more crashes in the zone, while "negative" means the opposite.

### 5.1. Built Environment and Public Transportation Stops

As shown in Figure 1a, almost all the zones have positive coefficients, which means that the more point of interest a zone contains, the higher the severity of bike crashes. Areas with a higher POI density may attract more cyclists, which can also contribute to the severity of bike crashes, which is similar to the findings of Hologa and Riach [69]. Regarding the traffic signals, it is noticed that the bike crash index is correlated positively in the inner areas of the city and correlated negatively in the suburbs, as shown in Figure 1b. Inner city areas may have higher traffic volumes and more complex road networks, which could lead to a higher incidence of bike crashes. In these areas, traffic signals may be more prevalent and may contribute to a higher number of bike crashes due to factors such as increased congestion and longer wait times at traffic signals. On the other hand, a negative correlation between bike crashes and traffic signals in the suburbs could indicate that the presence of traffic signals may be mitigating bike crashes in these areas. For example, traffic signals may be used to manage traffic flow and reduce speeds, which could make suburban roads safer for cyclists. These outcomes were also highlighted in Chen's (2015) [7] research. Similarly, as shown in Figure 1c, the relationship between bike crash index and road crossings is alike the traffic signals, which could be explained with the same approach. It should be noted that in all figures, the outputs are classified into four grades. We adopted the quantiles method from ArcGIS to grade the values.

Additionally, the results suggest that there is a spatially varying relationship between public transportation stops and bike crashes in the study area. Specifically, bus stops are more positively correlated with the bike crash index than railway/tram stops, as shown in Figure 1d. The bus stops are more numerous and more widely dispersed throughout the city than railway/tram stops, which could increase the likelihood of bike crashes occurring in their vicinity. The analysis has also revealed that the western and northern sides of the city are more highly influenced by bike crashes, due in part to the higher number of bus lines and stops in these areas. This suggests that there may be a need for targeted interventions to improve bike safety in these areas, such as the installation of dedicated bike lanes or the implementation of traffic-calming measures. For railway/tram stops, analysis has revealed that they are highly influenced by bike crashes in the southern side of the city as shown in Figure 1e. This may be due to the presence of rail/tram lines and stops in this area. In general, public transportation stops could be more frequently used

by cyclists as transfer points or as starting and ending points for bike trips, which could increase the risk of collisions with other vehicles or pedestrians. The results obtained in this research are alike the findings of Jaber and Csonka [54]. Finally, as shown in Figure 1f, bike lots are associated positively with the bike crash index. It was noted that the coefficient is especially high on the eastern side.

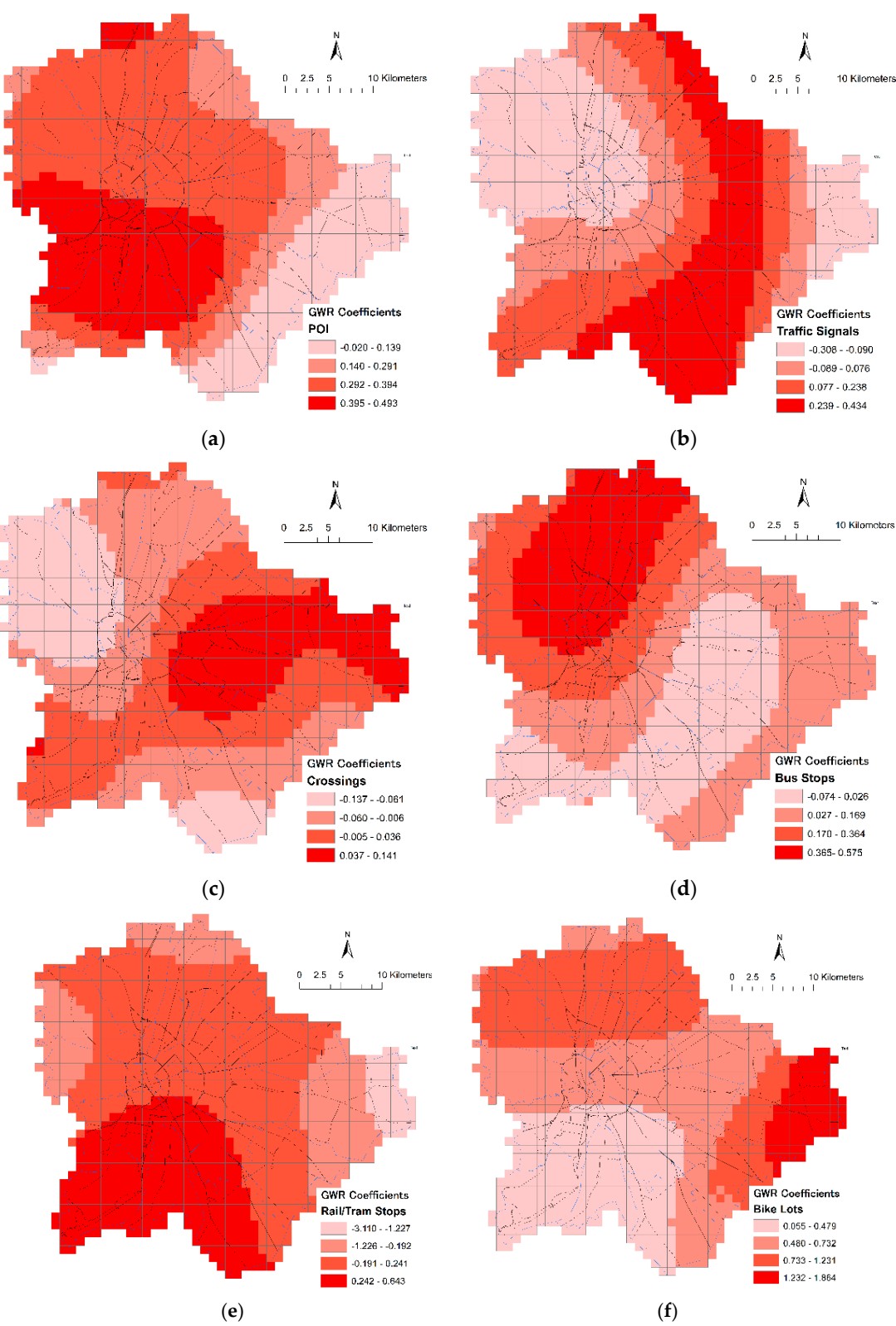

**Figure 1.** (**a**–**f**): GWR Coefficients of the Built Environment and Public Transportation Stops.

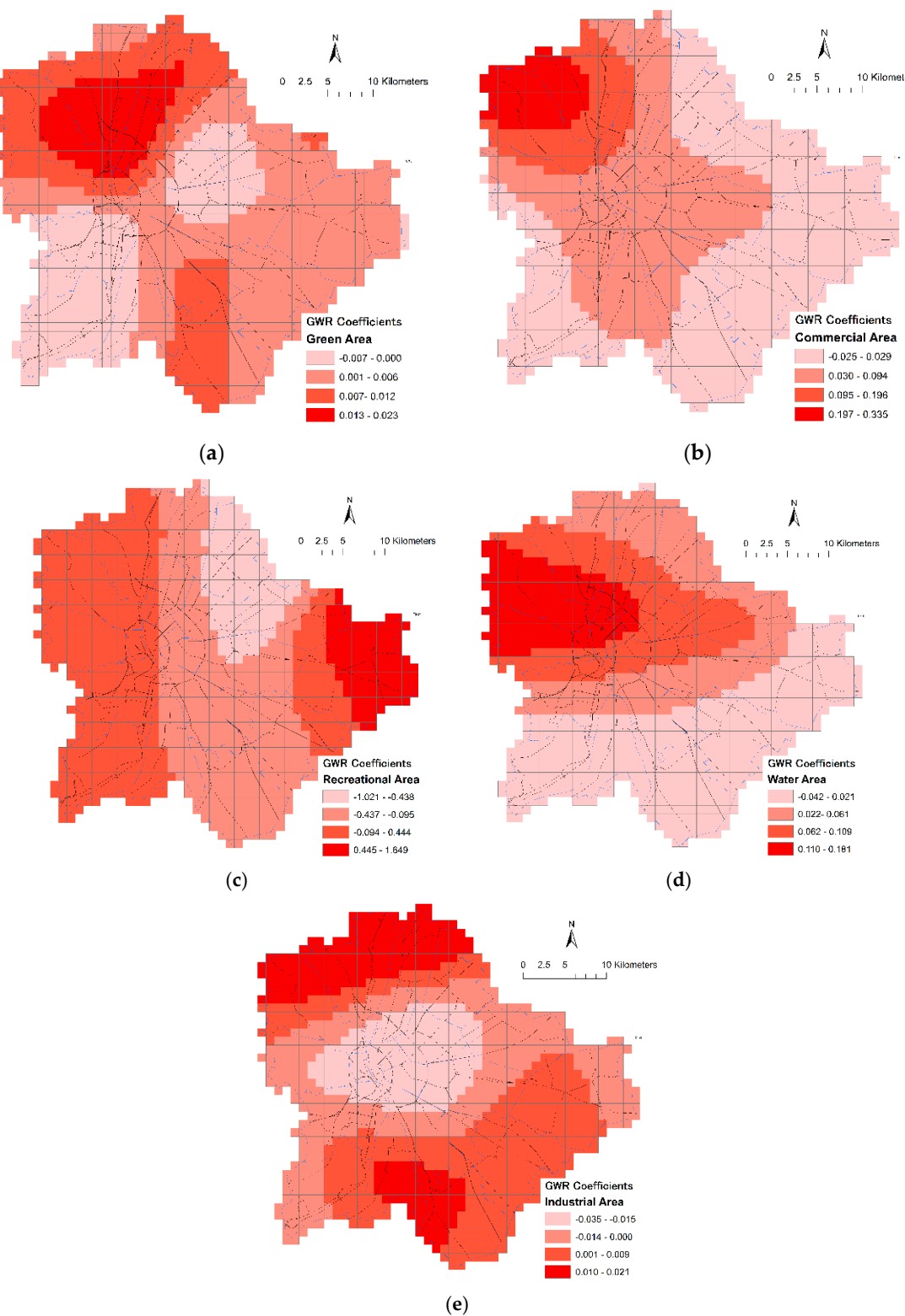

**Figure 2.** (**a**–**e**): GWR Coefficients of Land Use.

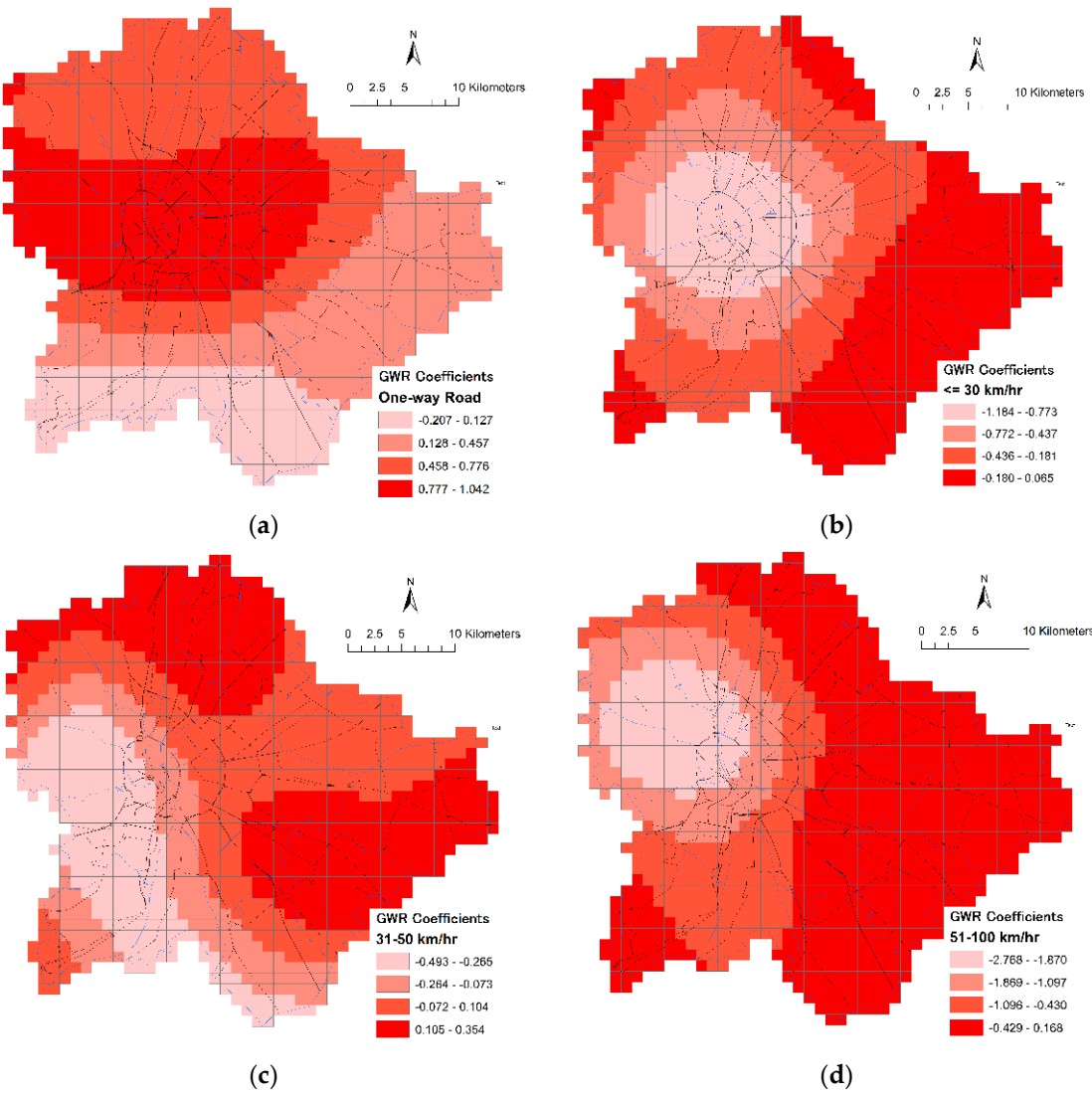

**Figure 3.** (**a**–**d**): GWR Coefficients of the Speed Limits and Road Direction.

### 5.2. Land Use

In Figure 2a, it is noticed that green areas have a low impact on the bike crash index as the coefficients are near to zero. This is an indication that the green areas are a safe environment for cyclists, which is stated in Panagopoulos et al.'s work [70]. Margit Island and the surroundings zones are the only zones that need attention because it is a very popular recreational area of the city. For the commercial areas (Figure 2b), it is clearly observed that the northern–western side of the city has the most impact on bike crashes as it also contains the main commercial activities in the city. For the recreational land-use areas, Figure 2c shows a negative correlation between land use and the bike crash index. This is aside from the green areas, which indicate a safe environment for cyclists in these zones. The only exception is the eastern side, as mentioned earlier, which has difficult topography that may affect severity. In aspects of the water area (Figure 2d), the bike crash index is correlated positively in Margit Island and the surrounding zone. Accordingly, this area needs some policies to mitigate bike crashes as far as possible. It is observed that water areas attract cyclists and pedestrians, which could increase the interaction between them and lead to more crashes. We are aware that there are few areas that are fully covered by water. These zones are taken into consideration by applying the bandwidth. Finally, as shown in Figure 2e, the industrial areas are shaped in a circular form towards the outside of the inner city.

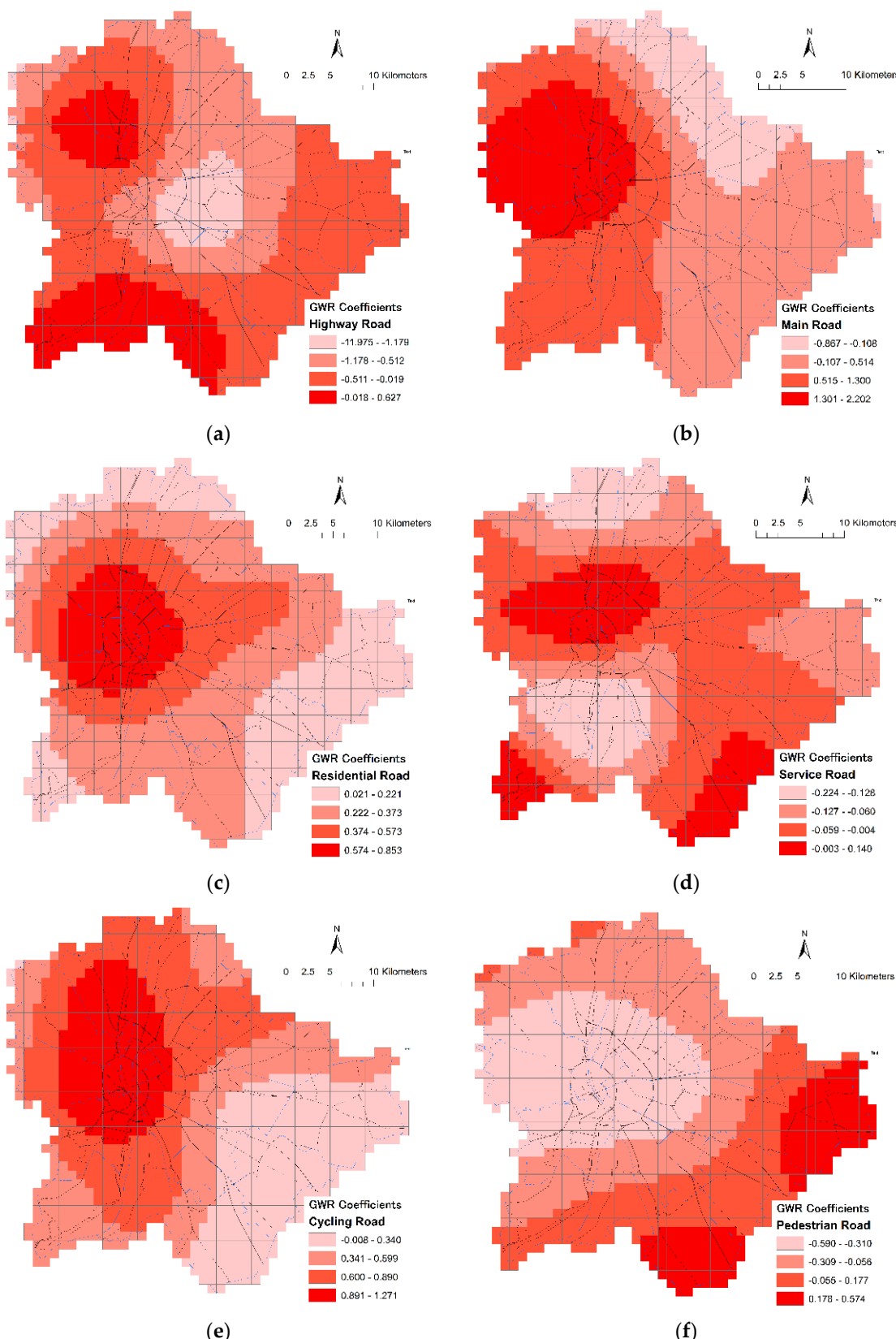

**Figure 4.** (**a**–**f**): GWR Coefficients of the Road Network Classification.

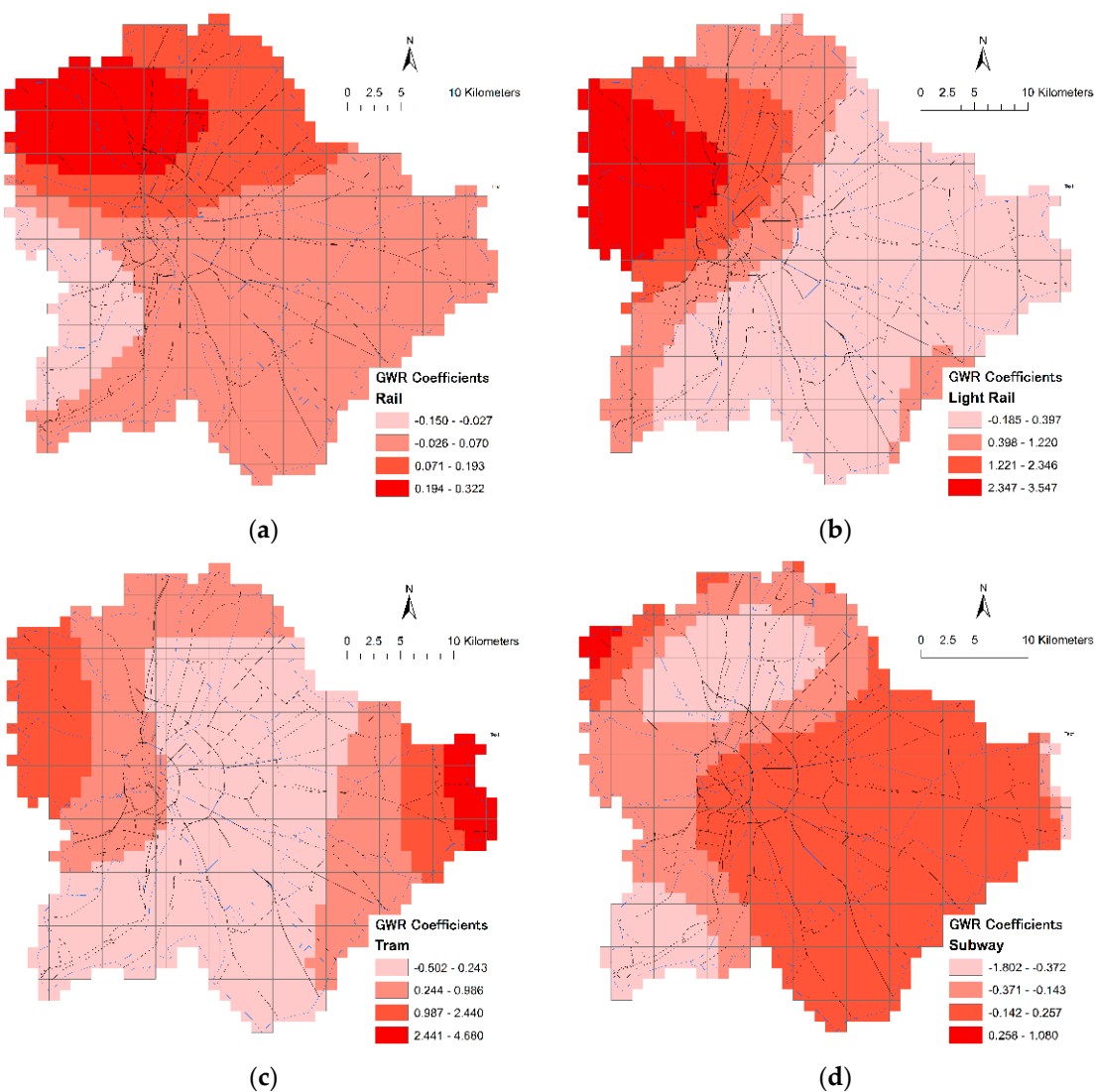

**Figure 5.** (**a**–**d**): GWR Coefficients of the Rail Network Classification.

### 5.3. Speed Limits and Road Direction

GWR analysis provided insights into the relationship between bike-crash-severity index and road characteristics, such as one-way roads and speed limits. Specifically, the analysis has shown that the more one-way roads an area has, the more severe bike crashes tend to be. One-way roads can increase the speed of vehicles and make it more difficult for cyclists to navigate; drivers also do not expect cyclists from both directions on a one-way road (many one-way roads are bidirectional for cyclists), which could contribute to the severity of bike crashes in these areas. These conclusions are partially on the contrary to of Raihan et al.'s (2019) [71] results, which showed a decreased number of crashes on one-way roads. In our research, the concentration of the coefficients is located in the center of the city as shown in Figure 3a, which is not fading towards outside. Moreover, the analysis has revealed that bike crashes in zones with speed limits of 30 km/h or lower are generally low, but they decrease towards the outer zones from the inner city as shown in Figure 3b. This could be due to factors such as lower traffic volumes and simpler road networks in the inner city, which could make it safer for cyclists to navigate. However, as cyclists move towards the outskirts of the city, traffic volumes and complexity of the road network may increase, which could increase the risk of bike crashes. Finally, the analysis has shown that areas with higher speed limits, particularly in suburban areas, are associated with a higher incidence of bike crashes, as shown in Figure 3c,d. These outcomes are similarly drawn in

Chen's (2015) [7] work. In Budapest, high speed limits are primarily found in outer zones, especially on the western side where the airport and regional roads are located.

*5.4. Road Network Classification*

GWR analysis also shed light on the relationship between bike crashes and various types of roads in Budapest. The results showed that the more of these roads there are, the more severe the bike crashes are, with the exception of pedestrian roads. Residential roads, main roads, and cycling roads all showed positive coefficients, indicating that they are correlated with higher severity of bike crashes as shown in Figure 4b, 4c, and 4e, respectively. This could be due to factors such as higher traffic volumes, more complex road networks, and faster speeds on these roads. In terms of residential roads, the GWR coefficients were concentrated in the inner and central zones of Budapest, suggesting that these areas may have a higher incidence of bike crashes on these types of roads. Similarly, the concentration of main roads in the inner and central zones also showed a positive correlation with bike crash severity. These results indicate that interventions may be needed to improve safety for cyclists on these types of roads, such as the creation of dedicated bike lanes or reducing speed limits.

Interestingly, the GWR coefficients for highway roads in the city center were negative, as shown in Figure 4a, indicating that there is almost no correlation between these types of roads and bike crashes in the inner city. This is likely due to the fact that there are very few highways in the city center, which could actually make it safer for cyclists. However, the results also showed that suburban areas with higher speed limits, particularly on the western side where several regional roads are located, had a higher incidence of bike crashes on highway roads. This suggests that further attention may be needed to improve bike safety on these roads outside the city center. For service roads, the coefficients vary spatially in negative and positive manners, as shown in Figure 4d. Finally, GWR analysis showed that pedestrian roads had a negative coefficient, as shown in Figure 4f, indicating that they are correlated with a lower severity of bike crashes. This is likely due to the fact that these roads are designed primarily for pedestrians and have lower traffic volumes, making them safer for cyclists as well. Overall, the findings suggest that interventions to improve bike safety should be tailored to the specific types of roads and areas where bike crashes are most severe.

*5.5. Rail Network Classification*

The findings suggest that the location of rail and light rail lines plays a significant role in the severity of bike crashes in Budapest. Even though railway services do not cover all the city, the coefficient maps cover the whole area, and due to that, these zones have been estimated by the bandwidth using the regression coefficients to consider the extent of the spatial autocorrelation. The coefficients of these factors are concentrated in the north and west side of the city as shown in Figure 5a,b, which is where the main rail stations are located. This could be attributed to the fact that the areas around the rail stations are usually busy with high traffic volumes, leading to a higher risk of bike crashes. Additionally, rail and light rail stations may attract a significant number of cyclists, who are more vulnerable to accidents in busy areas. On the other hand, the coefficients for tram lines and subways are distributed throughout the city, as shown in Figure 5c,d, reflecting the widespread distribution of these transportation modes in Budapest. This highlights the relationship between bike crashes and these transportation modes, as areas with a higher density of tram lines and subways may have more traffic and congestion, leading to an increased risk of bike crashes. However, the distribution of coefficients for these factors also suggests that bike crashes in proximity to tram lines and subways are not limited to specific areas, but rather occur throughout the city. Overall, the findings suggest that the relationship between transportation modes and bike crashes is complex and multifaceted, with different factors playing different roles in different areas of the city. Understanding these relationships can

help inform urban planning and transportation policies to reduce the risk of bike crashes and promote safe cycling.

An aggregation map showing the predicted values of GWR coefficients of the bike crash index can provide valuable insights into the factors that contribute to bike crashes in a particular area (Figure 6). This model output can help identify areas that are at higher risk of increased bike crash severity based on the factors described in the research. By comparing the aggregation map with the coefficient maps of each variable, we can identify which variables are the most significant contributors to bike crashes in that area. In this regard, the high-risk zones in inner areas of the city and the northern part are consistent with the coefficient maps of the light rail network, main roads, cycling roads, and one-way roads with approximate values of 1.37, 1.36, 0.94, and 0.91, respectively. However, the most influencing factors in the eastern–southern area are the trams network bike lots, recreational areas, and highway roads side with approximate coefficients of 2.21, 1.17, 0.77, and 0.30, respectively.

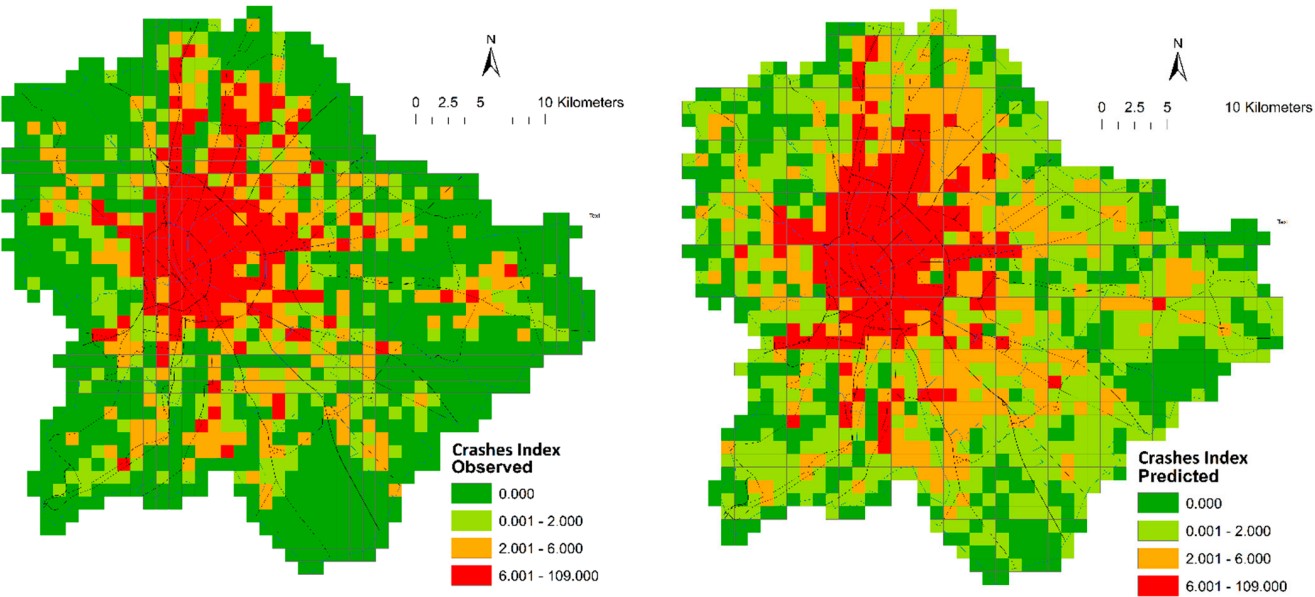

**Figure 6.** GWR Coefficients of the Observed and Predicted Model.

When comparing the observed bike crash index with our model, it is noticed that the GWR coefficients of the index are spatially almost similar, explaining 78.6% of the variation in the bike crash index by the independent variables.

In line with the growing recognition of streets as integral public spaces with significant impacts on urban livability and sustainability, our study finds resonance with the concept of "complete streets". As discussed by Montella et al. [72], this concept advocates for streets that cater to the needs of all users within a sustainable framework. Our research further aligns with the concept by introducing a bike-crash-severity index as a key component in understanding and enhancing cyclist safety. We draw inspiration from the sustainable complete streets design criteria presented in the mentioned source, which emphasizes the integration of socio-environmental design criteria related to aesthetics, environment, livability, and safety. While our study predominantly centers spatial analysis and risk assessment, we recognize the relevance of creating safe and sustainable urban environments for cyclists and all road users. This synergy underscores the significance of addressing cyclist safety within the larger context of urban development and sustainable transportation systems.

## 6. Conclusions

This study applied the geographic weighted regression (GWR) technique to investigate the spatial relationships between bike crashes and the built environment, land-use patterns, and transportation networks in Budapest, Hungary. The results showed that

certain variables, such as the presence of points of interest (POI) and traffic signals, have a significant impact on the frequency and severity of bike crashes in the city. The density of points of interest (POI) in a zone is positively correlated with bike crash severity. The relationship between bike crashes and traffic signals and road crossings is positive in inner areas of the city and negative in the suburbs. This may be due to the complexity of road networks and higher traffic volumes in inner areas, which could lead to more crashes. Bus stops are more positively correlated with bike crashes than railway/tram stops. The western and northern sides of the city are more influenced by bike crashes due to the higher number of bus lines and stops in these areas. Due to rail/tram lines and stops in the south, bike crashes are highly influenced. Green areas have a low impact on bike crash severity, indicating that they are safe environments for cyclists. Commercial areas have the most impact on bike crash severity in the northwestern side of the city. The more one-way roads an area has, the more severe bike crashes tend to be. Speed limits of 30 km/h or lower are associated with a lower severity of bike crashes. Bike lots are positively associated with bike crash severity, with a higher impact on the eastern side of the city. Additionally, the GWR approach allowed us to identify spatial heterogeneity in the relationships between these variables and bike crashes, which would have been missed by a global regression model. This information can be used to inform targeted interventions in areas with higher bike crash risk and to improve bike safety. The results of this study suggest that urban planners and policymakers should consider the spatial relationships between bike crashes and the built environment, land use, and transportation networks when developing strategies to improve bike safety and promote cycling.

This study suggests several bike-crash-prevention policies for Budapest. First, targeted interventions should be implemented in areas with a high bike-crash-severity index, such as the western–northern side of the city which are highly influenced by bike crashes due to the higher number of bus lines and stops in these areas. Traffic calming or bike lanes could reduce bike crashes in these areas. Additionally, Margit Island and surrounding zones, which are popular attraction areas of the city and have a high bike-crash-severity index, need policies to mitigate bike crashes as far as possible. For example, the installation of dedicated bike lanes or the implementation of speed limits could reduce the likelihood of bike crashes occurring in these areas. It is important to note that the coefficients of the statistically significant variables in the study exhibit heterogeneity, meaning that the relationships between the variables and bike-crash-severity index vary across the study area. For example, the relationship between traffic signals and bike crashes is positive in the inner areas of the city and negative in the suburbs (eastern–southern side). More consideration should be given to this side regarding the tram network and recreational areas. Therefore, it is necessary to take into account the spatial variability of the coefficients when developing policies to improve bike safety in the study area. Thus, the interventions should be implemented in areas with a high bike-crash-severity index and with a positive relationship between the variable and bike crashes, while areas with a negative relationship may need policies that maintain the existing infrastructure or expand it, with caution. The main limitation of our study is that we did not consider bike traffic volume or road intersections characteristics due to the lack of such data availability.

**Author Contributions:** Conceptualization, A.J. and B.C.; methodology, A.J.; software, A.J.; validation, A.J. and B.C.; formal analysis, A.J.; writing—original draft preparation, A.J.; writing—review and editing, A.J. and B.C.; visualization, A.J.; supervision, B.C. All authors have read and agreed to the published version of the manuscript.

**Funding:** This research received no external funding.

**Institutional Review Board Statement:** Ethical review and approval were waived for this study by the Vice Dean of the Transportation and Vehicle Engineering Faculty, the Discipline Committee of Budapest University of Technology and Economics, the Ethics Committee of Szeged University, and the Hungarian Medical Research Council that the research falls within a category where it was deemed exempt from requiring ethical approval. The study was conducted in accordance with the Declaration of Helsinki. Furthermore, and ethical approval for this study is not required in Hungary based on CLIVth Health Act of 1997—Chapter VIII, Section 157 since participants were neither subjected to interventions nor were they imposed by a code of conduct which would be protected by the Hungarian Medical Research Council. Data are treated confidentially, and individuals could not be identified from published data.

**Informed Consent Statement:** Not applicable.

**Data Availability Statement:** The data presented in this study are available on request from the corresponding author.

**Acknowledgments:** The authors would like to thank BKK for providing the data.

**Conflicts of Interest:** The authors declare no conflict of interest.

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
