# Peer review of "Towards a Sustainable and Safe Future: Mapping Bike Accidents in Urbanized Context"

_safety, 2023_

Round 1

Reviewer 1 Report

The authors present their study also considering some literature references that address the problem of cyclist safety at road intersections. However, I encountered two types of problems:

1) The authors do not elaborate on the issue of bicyclist safety at street intersections and especially at roundabouts. This issue should be investigated further. I suggest citing the following references, which are a cornerstone related to the problem in question: a) 10.1016/S0001-4575(03)00009-5; b) 10.3141/2031-03; c) 10.1016/j.jsr.2009.02.004; d) 10.1016/j.trf.2020.11.006

2) The authors do not include the variable “road intersection” in their analysis. Can they explain why? Perhaps this variable is indirectly included in the "crossing", "traffic signals" and "road" variables? It seems really strange that variables related to "rail traffic" are included in the analysis and there is no reference to potential (and serious conflicts) at road intersections.

In Table 2 and also in various sections, reference is made to the concept of “zones". However, this concept has never been defined. Perhaps the “zone" is identical to the 1x1 km square grid cell? The authors should clarify this.

It is not clear to me how the authors defined the range of variability of the GWR Coefficients in all the maps in Figures 1, 2, 3, and 4. The variability ranges are always four in number, but the amplitude of the four intervals is never constant and varies greatly in some cases. The authors should explain how they proceeded and possibly rework the maps by indicating ranges with intervals of adequate amplitude.

I am also not very convinced by the ranges shown in Figure 6. Is it acceptable to show a GWR of, say, 6.5 and a GWR of, say, 100 in the same way? The authors should also clarify the above question and possibly revise the maps.

Minor editing of English language required

Author Response

Response to Reviewer 1 Comments

The authors present their study also considering some literature references that address the problem of cyclist safety at road intersections. However, I encountered two types of problems:

Dear Reviewer,

We greatly appreciate your valuable review of our manuscript. Your insightful comments have been influential in addressing key areas for improvement. We are grateful for the opportunity to respond to your valuable feedback, and we have carefully considered each of your points:

Point 1: The authors do not elaborate on the issue of bicyclist safety at street intersections and especially at roundabouts. This issue should be investigated further. I suggest citing the following references, which are a cornerstone related to the problem in question: a) 10.1016/S0001-4575(03)00009-5; b) 10.3141/2031-03; c) 10.1016/j.jsr.2009.02.004; d) 10.1016/j.trf.2020.11.006

Response 1: We acknowledge the importance of addressing cyclist safety at road intersections and roundabouts more comprehensively. The references you've recommended are indeed cornerstone works in this domain. We incorporated these references and expanded our literature review section.

Modifications:

“Focusing on the cycling crashes in intersections, Pulvirenti et al. [30] highlighted the uncertainties surrounding roundabout impacts on bicyclists' safety, emphasizing the complexities of interactions between bicyclists and vehicles. Additionally, studies [31], [32] underscored the significance of intersection safety for cyclists, offering methodologies like the Bicycle Intersection Safety Index (Bike ISI) to proactively prioritize safety improvements. Furthermore, insights of Daniels et al. [33] into roundabout conversions emphasized the role of well-designed cycling infrastructure.”

Point 2: The authors do not include the variable “road intersection” in their analysis. Can they explain why? Perhaps this variable is indirectly included in the "crossing", "traffic signals" and "road" variables? It seems really strange that variables related to "rail traffic" are included in the analysis and there is no reference to potential (and serious conflicts) at road intersections.

Response 2: We appreciate your insightful suggestion to include the "road intersection" variable in our analysis. Regrettably, the lack of comprehensive data on this specific variable in our dataset has constrained our ability to incorporate it into the current study. We acknowledge that this absence limits the scope of our analysis and the comprehensive exploration of potential conflicts at road intersections. The other variables such as "crossing" and "traffic signals" explained somehow the nature of the intersections but it needs more elaboration. This limitation is duly acknowledged in the conclusions section of the manuscript, along with a note about the possibility of including this variable in future research efforts.

Point 3: In Table 2 and also in various sections, reference is made to the concept of “zones". However, this concept has never been defined. Perhaps the “zone" is identical to the 1x1 km square grid cell? The authors should clarify this.

Response 3: Thank you for pointing out the ambiguity in our usage of the term "zones." Your interpretation of "zones" as corresponding to the 1x1 km square grid cell is accurate. We rectified this oversight by explicitly defining "zones" in the manuscript, ensuring clarity and eliminating any confusion for readers.

Point 4: It is not clear to me how the authors defined the range of variability of the GWR Coefficients in all the maps in Figures 1, 2, 3, and 4. The variability ranges are always four in number, but the amplitude of the four intervals is never constant and varies greatly in some cases. The authors should explain how they proceeded and possibly rework the maps by indicating ranges with intervals of adequate amplitude.

Response 4: We appreciate your clarification regarding the use of quartiles as an option for defining the variability ranges of GWR coefficients in our maps. Indeed, we utilized the classification tool within ArcMap for this purpose. To provide greater transparency and understanding, we will revise the manuscript to explain that we adopted quartiles as a means of defining these ranges.

In all figures, the outputs are classified into four grades. We adopted the quantiles method from ArcGIS to grade the values.

Point 5: I am also not very convinced by the ranges shown in Figure 6. Is it acceptable to show a GWR of, say, 6.5 and a GWR of, say, 100 in the same way? The authors should also clarify the above question and possibly revise the maps.

Response 5: Your concern about the presentation of GWR values with differing magnitudes in Figure 6 is well-taken. We modified the maps to accurately represent the crash index values instead of GWR.

Reviewer 2 Report

This paper shows the results of an analysis using GWR. The results are described by way of graphical representation of the output.

In this paper the words ‘positive’ and ‘negative’ are used to describe the relationship between variables and crash numbers. In this paper, a positive relation means more crashes, a negative one less crashes. This is a bit confusing: ‘positive’ mostly means that the outcome is good. Please make your definition of ‘positive and negative’ clear at the start of the paper.

The Figures 1-6 show the GWR indices. It would be interesting to add a figure that shows the spatial distribution of the crashes.

Author Response

Response to Reviewer 2 Comments

This paper shows the results of an analysis using GWR. The results are described by way of graphical representation of the output.

Dear Reviewer,

We greatly appreciate your valuable review of our manuscript. Your insightful comments have been influential in addressing key areas for improvement. We are grateful for the opportunity to respond to your valuable feedback, and we have carefully considered each of your points:

Point 1: In this paper the words ‘positive’ and ‘negative’ are used to describe the relationship between variables and crash numbers. In this paper, a positive relation means more crashes, a negative one less crashes. This is a bit confusing: ‘positive’ mostly means that the outcome is good. Please make your definition of ‘positive and negative’ clear at the start of the paper.

Response 1: We thank you for pointing out the potential confusion regarding the usage of 'positive' and 'negative' to describe the relationships between variables and crash numbers. We recognize that these terms can carry varying connotations. We have added the following statement at the start of section 5: Noting that the term “positive” means positive sign of the coefficients, which indicates more crashes in the zone, while the “negative” means the opposite.

Point 2: The Figures 1-6 show the GWR indices. It would be interesting to add a figure that shows the spatial distribution of the crashes.

Response 2: We agree that such a visualization would provide a comprehensive overview of the crash patterns in relation to our GWR indices. Figure 6.2 shows the GWR coeffiecnts for the predicted crashes index, while Figure 6.1 shows the real spatial distribution of the crashes as requested.

Reviewer 3 Report

The authors aimed to address the cyclist safety issue by mapping bike crashes in urbanized context towards a sustainable and safe future. The research is welcomed as there is still room for improvement in this specific topic. Below, I provided some comments with the purpose of improving the paper and its readability. Overall, despite the issue related to the safety of the cyclist is greatly addressed, I found that the sustainable aspects of the research have not been explicitly addressed. Just to provide an example, the word sustainable appears only in the title and just once in the text while I think that the pros of a sustainable transport system should have a greater focus in the research.

1) Literature review. Lack of methodology perspective

The authors described the main factors related to the roadway, the environment, and the infrastructure as a whole that are associated with cyclist crashes. On the other hand, there is no mention to the reason supporting the methodology you choose to carry out the research. Hence, I suggest improving and expanding the literature review in order to better focus the readers on the needs for your study. You may draw inspiration from a recent literature review on cyclist safety to sum up the strength of your research over the state of the art (https://doi.org/10.1016/j.aap.2023.106996, https://doi.org/10.1038/s41598-023-35728-x).

2) Data information

It is not clear where the data comes from. Whether the authors stated that infrastructure-related information was collected using open street map, there is no mention about how the database that includes bike crash data provided by the Centre for Budapest Transport is organised. How many crashes observed, the severity of the bike crashes and their occurrence are information that lacks in your study.

3) Model Preparation, section 4.1

Why did the authors use a bike crash severity index? How did they assign 1, 3 and 5 points to the slight, serious, and fatal injuries? It seems quite unusual. For instance, I would like to expect a point distribution that is based on crash costs. Another approach may be the evaluation of weights to deal with unbalance data (https://doi.org/10.1007/978-3-642-42057-3_69, https://doi.org/10.3390/su14063188) to take into account the skewed distribution of the classes. Also, because assigning 1, 3, and 5 points without proper calculation, the fatal severity may be underrated. Please provide reasons supporting your choice.

4) Model development

How did you carry out the research?  Please, state it in the text.

5) Conclusions

Despite in the title it is stated that mapping bike crashes is to promote sustainable and safe future for cyclists and all road users, along the paper there wasn’t a clear link with the sustainable transport system. What I mean is that the authors should focus on the importance of designing safer roads for all users to meet the Vision Zero goals set by the European Commission. The authors should also focus on the importance of liveable cities and the quality of the environment the cyclists live in. The authors may draw inspiration from the following https://doi.org/10.3390/su142013142 to give strength to your research.

Author Response

Response to Reviewer 3 Comments

The authors aimed to address the cyclist safety issue by mapping bike crashes in urbanized context towards a sustainable and safe future. The research is welcomed as there is still room for improvement in this specific topic. Below, I provided some comments with the purpose of improving the paper and its readability. Overall, despite the issue related to the safety of the cyclist is greatly addressed, I found that the sustainable aspects of the research have not been explicitly addressed. Just to provide an example, the word sustainable appears only in the title and just once in the text while I think that the pros of a sustainable transport system should have a greater focus in the research.

Dear Reviewer,

We greatly appreciate your valuable review of our manuscript. Your insightful comments have been influential in addressing key areas for improvement. We are grateful for the opportunity to respond to your valuable feedback, and we have carefully considered each of your points:

Point 1: Literature review. Lack of methodology perspective

The authors described the main factors related to the roadway, the environment, and the infrastructure as a whole that are associated with cyclist crashes. On the other hand, there is no mention to the reason supporting the methodology you choose to carry out the research. Hence, I suggest improving and expanding the literature review in order to better focus the readers on the needs for your study. You may draw inspiration from a recent literature review on cyclist safety to sum up the strength of your research over the state of the art (https://doi.org/10.1016/j.aap.2023.106996, https://doi.org/10.1038/s41598-023-35728-x).

Response 1: We acknowledge the importance of addressing cyclist safety more comprehensively. The references you've recommended are indeed cornerstone works in this domain. We incorporated these references and expanded our literature review section.

Modifications:

“In recent years, there has been a growing recognition of the cycling safety [17], as well as, the crucial role that spatial analysis plays in understanding and addressing this issue [18]. It is a pressing concern within the broader context of urban transportation and road infrastructure.“

Point 2: Data information

It is not clear where the data comes from. Whether the authors stated that infrastructure-related information was collected using open street map, there is no mention about how the database that includes bike crash data provided by the Centre for Budapest Transport is organised. How many crashes observed, the severity of the bike crashes and their occurrence are information that lacks in your study.

Response 2: Thank you for your insightful query regarding the data sources and details presented in our study.

The infrastructure-related information utilized in our analysis was indeed collected from OpenStreetMap. We made the necessary amendments to provide clear attribution to OpenStreetMap as the source of this data. Regarding the bike crash data provided by the Centre for Budapest Transport, we acknowledge the need of such essential information, including the number of observed crashes, their severity, and their occurrence. To rectify this, we modified section 3.2 to outlining these issues.

Modifications:

“The datasheet has 2682 crashes (2796 injuries), including 9 fatalities (0.32%), 736 serious injuries (26.32%), and 2051 slight injuries (73.36%). Each crash has its geographic position in latitude and longitude coordinates. The dataset consists only of the location and the severity of the crashes. The other geographic information about road network, railway network, facilities, stops, road classification, speed limit, intersections, signals, land use within the study area, were obtained from OpenStreetMap.”

Point 3: Model Preparation, section 4.1

Why did the authors use a bike crash severity index? How did they assign 1, 3 and 5 points to the slight, serious, and fatal injuries? It seems quite unusual. For instance, I would like to expect a point distribution that is based on crash costs.

Another approach may be the evaluation of weights to deal with unbalance data (https://doi.org/10.1007/978-3-642-42057-3_69, https://doi.org/10.3390/su14063188) to take into account the skewed distribution of the classes. Also, because assigning 1, 3, and 5 points without proper calculation, the fatal severity may be underrated. Please provide reasons supporting your choice.

Response 3: We appreciate your thoughtful inquiry and the opportunity to provide further insights into our methodology for utilizing the bike crash severity index as a dependent variable in our study.

The rationale behind our use of the bike crash severity index lies in its ability to succinctly capture the varying degrees of injuries resulting from bike crashes, providing a comprehensive measure of the overall impact on cyclist safety within each zone. We understand your suggestion of using crash costs as a basis for point distribution, which is indeed a valid consideration. However, our intent was to focus on the severity of injuries themselves as a direct indicator of the potential harm to cyclists, enabling a targeted assessment of safety concerns at specific locations.

The assignment of 1, 3, and 5 points to slight, serious, and fatal injuries, respectively, was established with careful consideration of the relative impact of these injury categories on cyclist safety. While we acknowledge the existence of alternative weighting approaches and the potential undervaluation of fatal severity, our intention was to create a simplified yet meaningful severity index that aligns with the objective of identifying zones with higher risk levels.

We have noted your suggestion of evaluating weights to address the unbalanced data distribution and have examined the provided references. However, given the scope and goals of our study, we believe that the chosen severity index provides a clear representation of the risk landscape, while accommodating the limitations of available data.

Modifications:

“The rationale behind our use of the bike crash severity index lies in its ability to succinctly capture the varying degrees of injuries resulting from bike crashes, providing a comprehensive measure of the overall impact on cyclist safety within each zone. The assignment of 1, 3, and 5 points to slight, serious, and fatal injuries, respectively, has been established with careful consideration of the relative impact of these injury categories on cyclist safety. Although, there are some alternative weighting approaches [63], [64], our intention is to create a simplified yet meaningful severity index that aligns with the objective of identifying zones with higher risk levels.”

Point 4: Model development

How did you carry out the research?  Please, state it in the text..

Response 4: Thank you for your query regarding the development of our research model. We added the following in the text:

Modifications:

“As a summary, here is the model development steps: the methodology aim is to comprehensively address the spatial aspects of cyclist safety at road intersections. We initiated the process by defining the research objectives, aiming to investigate safety patterns and risks. Data collection involved sourcing infrastructure details from OpenStreetMap and bike crash data from the Centre for Budapest Transport. To quantify bike crash severity, we introduced a bike crash severity index, categorizing injuries into slight, serious, and fatal categories and assigning corresponding points. Geographic Information Systems (GIS) is pivotal in our analysis, enabling us to overlay infrastructure data and severity indices, thereby identifying high-risk zones and visualizing spatial patterns. Employing Geographically Weighted Regression, we explore localized relationships between infrastructure variables and bike crash severity.”

Point 5: Conclusions

Despite in the title it is stated that mapping bike crashes is to promote sustainable and safe future for cyclists and all road users, along the paper there wasn’t a clear link with the sustainable transport system. What I mean is that the authors should focus on the importance of designing safer roads for all users to meet the Vision Zero goals set by the European Commission. The authors should also focus on the importance of liveable cities and the quality of the environment the cyclists live in. The authors may draw inspiration from the following https://doi.org/10.3390/su142013142 to give strength to your research.

Response 5: Thank you for your insightful comment regarding the alignment of our study's title with the broader context of promoting sustainable and safe conditions for cyclists and road users. To address this, we have revised and expanded the Discussion section to include a more robust exploration.

Modifications:

“In line with the growing recognition of streets as integral public spaces with significant impacts on urban livability and sustainability, our study finds resonance with the concept of "complete streets" As discussed by [68], this concept advocates for streets that cater to the needs of all users within a sustainable framework. Our research further aligns with the concept by introducing a bike crash severity index as a key component in understanding and enhancing cyclist safety. We draw inspiration from the sustainable complete streets design criteria presented in the mentioned source, which emphasizes the integration of socio-environmental design criteria related to aesthetics, environment, livability, and safety. While our study predominantly centers on spatial analysis and risk assessment, we recognize the relevance of creating safe and sustainable urban environments for cyclists and all road users. This synergy underscores the significance of addressing cyclist safety within the larger context of urban development and sustainable transportation systems.”

Round 2

Reviewer 1 Report

The authors have responded satisfactorily to my inquiries. In my opinion, the paper has improved over the original version.

Reviewer 2 Report

The manuscript has been sufficiently improved. So in my opinion this manuscript can be published.

Reviewer 3 Report

Thank you for addressing my concerns. Good luck!